# Variable Expression of GABAA Receptor Subunit Gamma 2 Mutation in a Nuclear Family Displaying Developmental and Encephalopathic Phenotype

**DOI:** 10.3390/ijms23179683

**Published:** 2022-08-26

**Authors:** Gerald Nwosu, Shilpa B. Reddy, Heather Rose Mead Riordan, Jing-Qiong Kang

**Affiliations:** 1Department of Neuroscience and Pharmacology, Meharry Medical College, Nashville, TN 37208, USA; 2Department of Neurology, Vanderbilt University Medical Center, Nashville, TN 37232, USA; 3Brain Institute, Vanderbilt University, Nashville, TN 37232, USA; 4Department of Pediatrics, Vanderbilt University Medical Center, Nashville, TN 37212, USA; 5Department of Pharmacology, Vanderbilt University, Nashville, TN 37233, USA

**Keywords:** epilepsy, developmental epileptic encephalopathies, GABA_A_ receptor, mutations, gene therapy, antisense oligonucleotides

## Abstract

Mutations in GABA_A_ receptor subunit genes (*GABRs*) are a major etiology for developmental and epileptic encephalopathies (DEEs). This article reports a case of a genetic abnormality in *GABRG2* and updates the pathophysiology and treatment development for mutations in DEEs based on recent advances. Mutations in *GABRs*, especially in *GABRA1*, *GABRB2*, *GABRB3*, and *GABRG2*, impair GABAergic signaling and are frequently associated with DEEs such as Dravet syndrome and Lennox–Gastaut syndrome, as GABAergic signaling is critical for early brain development. We here present a novel association of a microdeletion of *GABRG2* with a diagnosed DEE phenotype. We characterized the clinical phenotype and underlying mechanisms, including molecular genetics, EEGs, and MRI. We then compiled an update of molecular mechanisms of *GABR* mutations, especially the mutations in *GABRB3* and *GABRG2* attributed to DEEs. Genetic therapy is also discussed as a new avenue for treatment of DEEs through employing antisense oligonucleotide techniques. There is an urgent need to define treatment targets and explore new treatment paradigms for the DEEs, as early deployment could alleviate long-term disabilities and improve quality of life for patients. This study highlights biomolecular targets for future therapeutic interventions, including via both pharmacological and genetic approaches.

## 1. Introduction

Developmental and epileptic encephalopathies (DEEs) are a collection of disorders and syndromes defined by an underlying genetic cause, which results in both a neurodevelopmental and epileptic phenotype. Due to early onset, the progression of epileptic encephalopathy conjoins to dampen normal maturation of the brain throughout development and continues to impact brain function [1,2]. The resultant phenotype manifests as unremitting and often pharmaco-resistant seizures, behavioral dysfunction, neurodevelopmental delay, and lowered quality of life for the patient. There are a host of genes that have been identified and, when mutated, are causative for the developmental and epileptic encephalopathic phenotype. These gene mutations affect brain function through altering GABAergic signaling, GABA transport, ion channel function, and cortical formation among other mechanisms [3,4,5,6,7,8,9].

GABA_A_ receptor genes (*GABR*) are a group of genes associated with DEE, although previous studies have been focused on relatively mild epileptic syndromes, such as childhood absence epilepsy (CAE), febrile seizures (FS), and generalized tonic-clonic seizures with febrile seizure plus (GEFS+). DEEs such as Dravet syndrome (DS), Lennox–Gastaut syndrome (LGS), West syndrome (WS), or infantile spasm (IS) commonly show epileptic activity comorbid with intellectual disability or autism spectrum disorder and have been frequently associated with mutations in *GABRs*. Some patients with *GABR* variants present with neurodevelopmental delay or impaired cognition without seizures, indicating that seizure is just one part of disease phenotype.

GABA signaling is fundamental for normal brain development and function. The GABA_A_ receptors are responsible for both phasic and tonic inhibitory postsynaptic potentials that dictate neuronal communication in the central nervous system [10,11]. The GABA_A_ receptors are members of a super family of Cys-loop, ligand-gated ion channels possessing a characteristic loop formed by a disulfide bond between two cysteine residues [12]. GABA_A_ receptors are composed of three of nineteen subunits in the family, present at the cell surface and synapse as a hetero-pentamer. There have been at least ten GABA_A_ receptor subunit genes (GABR) associated with various genetic epilepsy syndromes. Such genes include *GABRA1-3*, *GABRA5-6*, *GABRB1-3*, *GABRG2*, and *GABRD* [13,14,15]. The clinical diagnoses associated with GABR mutations include CAE, FS, juvenile myoclonic epilepsy (JME), GEFS+, DS, and LGS [16,17,18,19,20,21,22]. We have identified multiple mechanisms affecting receptor biogenesis and trafficking, ranging from reducing subunit mRNA transcription or stability; impairing subunit folding, stability, or oligomerization; or inhibiting receptor trafficking [14,18,23,24]. Thus, there are various mechanisms underlying *GABR* mutations associated with DEEs.

Here, we report a novel case of DEE associated with a *GABRG2* mutation and compiled the mechanisms of recent studies with a focus on both *GABRG2* mutations and *GABRB3* mutations. We focus on these two subunits because mutations in *GABRG2* and *GABRB3* are more likely to display common phenotypical characteristics. We thus compiled the perturbed biomolecular mechanisms of *GABRB3* and *GABRG2* mutations and grouped them based on affected function for design and application of precise therapeutic intervention.

## 2. A Novel Case of *GABRG2* Mutation Associated with DEE, Expanding the Disease Phenotypes Associated with *GABRG2* in DEEs

Recently, a nine-year-old male child was found to have a *GABRG2* mutation. Sequence analysis and deletion/duplication testing of the 151 genes in the Invitae Epilepsy Panel were performed. This showed there was a heterozygous deletion of the entire coding sequence for the *GABRG2* gene, which was also found in his asymptomatic mother and twin brother, who only has a history of febrile seizures. In this case, it will be useful to obtain a genotype of *GABRG2* in the proband and twin brother to further define the phenotypic spectrum of *GABRG2* mutations. This could delineate the difference in symptom severity. The child has a history of moderate global developmental delay in addition to intractable symptomatic generalized epilepsy. It is intriguing that the mother, who is carrier of the mutation, is asymptomatic, while the monochorionic-monoamniotic twin brother only has febrile seizures (Figure 1A). This is in corroboration with many reports of a male predominance to the DEE phenotype associated with mutations in genes such as CDKL5 and SCN1A. It is also possible that the mother is only a carrier, with carrying males expressing the phenotype [3,25]. *GABRG2* encodes the γ_2_ subunit of GABA_A_ receptor, one of the most abundant subunits critical for clustering the pentameric receptors at the synapse (Figure 1B). The patient showed generalized-appearing spike and polyspike wave discharges (Figure 1C) and multifocal epileptiform discharges, most prominent in the right mesial parietal (Figure 1D) and left posterior regions (Figure 1E). Similar intrafamilial phenotypic heterogeneity has been observed in other *GABRG2* mutations. Of note, the proband had a history of hepatoblastoma treated with chemotherapy and resection; thus, one possible explanation for the phenotypic difference could be changes in brain structure and function related to the chemotherapy, resulting in the more severe seizure semiology [26]. This suggests that an underlying condition could modify disease phenotype presentation during development.

The proband’s first seizure was reported at the age of four, presenting as an unprovoked focal tonic seizure with shifting gaze deviation. At this time, the EEG showed interictal sharp waves in the right-frontal region. Magnetic resonance imaging showed multifocal abnormality with abnormal configuration of lateral and third ventricle (Figure 2A,B). This suggests minimal infra- and supratentorial volume loss, microcephaly, abnormal folia architecture within the cerebellum, and abnormal appearance of the globes. These findings are nonspecific and may be suggestive of the underlying genetic mutation. Possible causes would include variable penetrance, modifier genes, or epigenetic/environmental factors. The etiology of phenotype can ultimately determine the treatment paradigm for patients.

He first walked at the age of four and continues to walk on his toes with a wide-based gait. The proband was diagnosed with autism spectrum disorder (ASD) at the age of six. He continues to have global delays. Seizures occurred both during wakefulness and sleep, with all EEG recordings being obtained around nine years of age. At the time of his last visit when he was at age ten and no longer experienced myoclonic or atonic seizures but continued to have non-epileptic myoclonus multiple times per day and generalized seizures (characterized by a slow head drop and behavioral arrest) three times per week. He has frequent complex motor stereotypies. His exam is otherwise notable for subtle ataxia and mild contractures in his gastrocnemius muscles, felt to be related to idiopathic toe walking as opposed to underlying spasticity. The patient was started on levetiracetam, but at age five, seizures became more frequent with multiple semiologies. The background activity was consistent with multifocal and generalized potential epileptogenicity and generalized cerebral dysfunction, including myoclonic and atonic seizures (Figure 1C–E, Appendix A) as well as decreased responsiveness associated with quasi-rhythmic generalized discharges lasting ten seconds to twenty minutes. Seizures continued despite escalation of antiepileptic medications, with lacosamide and levetiracetam proving ineffective. Topiramate and valproic acid were discontinued due to side effects, including decreased clearance of clobazam, leading to toxicity. At the age of nine, he developed multifocal, non-epileptic myoclonus. This corresponded with the cessation of topiramate, and he did not respond to benzodiazepines. His current antiepileptic regimen consists of rufinamide, clobazam, and epidiolex.

## 3. Unique Functional Properties of *GABRG2*-Encoded γ_2_ Subunit

As mentioned above, the *GABRG2*-encoded γ_2_ subunit is a member of the GABA_A_ receptor subunit family that plays a unique role in postsynaptic clustering of the receptor during synaptogenesis [27]. The majority of GABA_A_ receptors found at postsynaptic sites contain the γ_2_ subunit and are essential for the clustering of α_1_ and α_2_ receptors at such sites via the scaffold protein gephyrin [28]. It has been demonstrated that the γ_2_ subunit is responsible for the accumulation of gephyrin, which in turn is responsible for the synaptic clustering of major subtypes of the GABA_A_ receptor [29,30]. Thus, the γ_2_ subunit is required for normal channel function and for postsynaptic clustering of the receptor during synaptogenesis [30]. Regulation of proper synaptic localization is a contributing factor of functional plasticity at excitatory and inhibitory synapses. The γ_2_ subunit containing ternary receptors confers larger channel conductance than the binary αβ receptors [31]. It has been previously reported that mice deficient in the γ_2_ subunit show a reduction in channel conductance, open probability, and open duration [32,33]. The γ_2_ subunit is also essential for the formation of a benzodiazepine binding site, which, when disrupted, can result in pharmaco-resistance to certain drugs, as seen in the patient detailed above [33]. In summary, there are at least two unique features for the γ_2_ subunit. First, it is essential for channel opening with a larger conductance compared with other GABA_A_ receptor subunits. Second, the γ_2_ subunit is a requirement for synaptic localization and clustering of GABA_A_ receptors during development and maturation, thereby being essential for the proper facilitation of inhibitory postsynaptic signaling. Considering the critical role of the γ_2_ subunit in synaptic GABA_A_ receptor function, mutations of this gene can lead to altered GABA signaling and GABAergic neurotransmission, a critical pathomechanism underlying the clinical DEE phenotype.

## 4. Pathomechanisms for *GABRG2* Mutations in DEEs

The most common phenotypes associated with *GABRG2* mutations are febrile seizures, generalized tonic-clonic seizures with febrile plus (GEFS+), and DS, with febrile seizures being the core phenotype (Figure 3A). Extensive work from previous studies has shown that *GABRG2* mutations are associated with a wide spectrum of seizure disorders (Figure 3B). There has been much investigation into the impact of *GABRG2* mutations in vitro and in vivo using knock-in mouse models, which would allow us to infer the pathophysiology of this mutation in the patient.

Mutations of *GABRG2* are frequently associated with developmental and epileptic disorders through both missense and nonsense mutations. Nonsense mutations in *GABRG2* have been identified and are associated with FS, GEFS+, or DEEs. A *GABRG2* (*R43Q*) mutation had reduced cell surface expression of the subunit and reduced cortical inhibition in knock-in mice [34,35]. However, in some cases those mutant receptors, which traffic to the cell surface, have normal function [36]. It was also identified that a subset of mutations, especially the nonsense mutations *GABRG2* (Q351X), cannot traffic to the cell surface and synapse due to severe misfolding [37,38]. The mutant protein was retained in the endoplasmic reticulum and reduced trafficking of wildtype partnering subunits, implicating its dominant-negative suppression causing GEFS+. The mutant *GABRG2* (Q351X) has slow degradation and can impose a dominant-negative effect on the wildtype *GABRG2* allele and the partnering subunits [37]. These findings have been validated in vivo with *Gabrg2^+/Q390X^* mice, showing that the mutation *GABRG2* (Q390X) is proven to cause chronic subunit accumulation and neurodegeneration [38]. In older mice, the mutated subunit was shown to form protein aggregates due to chronic accumulation of the mutant protein in the endoplasmic reticulum [38]. Mice expressing the *GABRG2* (Q390X) mutation showed a severe epileptic phenotype, which included spontaneous generalized tonic-clonic seizures in a seizure-resistant C57BL/6J mouse background. The phenotype is much more severe than the *Gabrg2^+/−^* heterozygous knockout mice without generation of the mutant protein [39].

As mentioned above, the γ_2_ subunit plays an essential role in GABAergic neurotransmission and the overall homeostasis of the central nervous system, while impairment in the subunit results in seizures and neurodevelopmental abnormalities. Some mutants of this protein display a dominant-negative effect that suppresses expression of both wildtype subunit and its binding partners in the hetero-pentamer. Of note, the proband displays an uncharacteristic phenotype for a mutation of this subunit, suggesting further investigation into the interplay of associated proteins with the receptor and how their function is affected by the mutation. The GABA_A_ receptor exerts channel function as a pentamer, and the γ_2_ subunits oligomerize and colocalize with partnering subunits. There is the possibility that the deficiency due to mutant γ_2_ protein can result in suppression of proper function of the β_3_ subunit whose perturbation is often attributed to the LGS phenotype.

## 5. Unique Functional Properties of *GABRB3* Encoded β_3_ Subunit

*GABRB3* has been identified as an emerging cause for early infantile developmental and epileptic encephalopathies. This subunit is highly expressed in the embryonic brain suggestive of its role in neuronal migration, synaptogenesis, and brain development and maintains expression in the hippocampus exclusively during development [40]. It has come to consensus in the field that the β_3_ subunit is a major subunit seen in the majority of synaptic GABA_A_ receptors. Like other GABA_A_ receptor subunits, the β_3_ subunit is composed of a large extracellular NH_2_ terminal region, four hydrophobic transmembrane domains, an M3-M4 linker, and a short carboxyl terminal. The M2 segment is exposed to the ion channel pore along with the other four subunits of the receptor [41,42]. This is important, as the position of the mutation can affect ion channel capabilities for the whole receptor. *GABRB3* has a regulatory role in membrane insertion, as it contains specific endocytosis motifs within the intracellular domains for the clathrin adaptor protein *AP2*. It is worth noting that the β_3_ subunit protein contains two phosphorylation sites in its TM3-TM4 intracellular loop. Specifically, phosphorylation of the β_3_ subunit at S408/S409 is significant for regulating GABA_A_ receptor endocytosis [43]. *GABRB3* can itself impose direct impact on channel function, and genetic abnormalities in other genes can also alter *GABRB3* expression and contribute to disease pathophysiology. For example, reduced *GABRB3* gene expression has been hypothesized as the pathogenesis of Rett syndrome (RS), Angelman syndrome (AS), ASD, and LGS [44,45]. There is an important structural role for *GABRB3* akin to *GABRG2*, so the pathophysiology and phenotypes associated with *GABRG2* and *GABRB3* mutation may overlap.

## 6. Pathomechanisms for *GABRB3* Mutations in DEEs

Like the phenotypic heterogeneity associated with mutations in *GABRG2*, mutations in *GABRB3* have also been associated with a spectrum of disease phenotypes, including ASD, IS, and DS [46,47] (Figure 4A,B). Among all *GABRs*, only mutations in *GABRB3* have been previously reported to be associated with LGS [20]. In addition to LGS, the β_3_ subunit encoded by *GABRB3* has also been frequently associated with CAE [32,48]. In addition to the defect in the mutant β_3_ subunit, a mutant β_3_ subunit can also compromise the assembly and trafficking of partnering subunits α_1_ or γ_2_. This suggests there may exist overlapping phenotypes at the clinical level among mutations in the partnering subunits of the receptor through a dominant-negative effect of one, suppressing the maturation and trafficking of the others, as previously reported [39].

Mutations in *GABRB3* have been frequently related to DEEs (Figure 4). A previous study evaluated the mutations identified by the Epilepsy Phenome/Genome Project (D120N, E180G, and Y302C) [46]. Follow-up functional assays identified the alterations of GABA-activated channel function, including reduced GABA-evoked current amplitudes, slowed activation, and accelerated deactivation [49]. This was inferred to be due to a reduction in the GABA potency, representing the concentration of GABA, and efficacy, representing the potential of GABA to bind, caused by the mutations. Specifically, the D120N mutation reduced GABA potency, whereas the E180G and Y302C mutations reduced GABA efficacy [49]. This implicates that different mutations can have separate effects on the inhibitory neurotransmission mechanisms. It is important to note that each mutation was in either loop A or loop B of the GABA-binding pocket or the M2-M3 loop, which is involved in the ligand-binding channel, gating-coupling mechanism. The specific pathomechanism could be the likely cause of the mutations disrupting coupling of GABA binding to channel gating.

The differential pathomechanisms in *GABRB3* mutations associated with epilepsy with variable severities have been identified [50]. There was a thorough comparison of the *GABRB3* (N328D) mutation associated with LGS with *GABRB3* (E357K), a mutation associated with a less severe phenotype: juvenile absence epilepsy [50]. In the patient carrying the N328D mutation, there was presence of generalized tonic-clonic seizures and myoclonic seizures. The initial EEG examination showed ictal 2–2.5 Hz generalized polyspike-wave discharges during myoclonic seizures, interictal spike, and slow wave complexes, and irregular fast rhythms with slow waves were recorded during wakefulness [50]. It was identified that the LGS-associated *GABRB3* (N328D) mutation caused more reduced cell-surface expression and synaptic presentation of α_1_β_3_γ_2_ than *GABRB3(E357K)* mutation associated with the juvenile absence epilepsy. However, both mutations caused trafficking defects due to endoplasmic reticulum retention of the mutant protein. Through use of a high-throughput flow cytometry assay, the surface expression of β_3_ and γ_2_ subunits for several *GABRB3* mutations was evaluated. There was a significant reduction in the expression of the β_3_ and γ_2_ subunits at the cell surface for both mutations [50]. This implies that a mutation of this subunit not only prevents proper trafficking of the individual subunit itself but for the partnering wildtype subunits such as γ_2_ and α_1_ subunit expression, suggesting loss of whole receptor function instead of subunit alone. By comparing mutations in *GABRB3* associated with different phenotypes or different mutations with the same phenotype such as LGS, it is likely that different mutations can have separate effects on the inhibitory neurotransmission mechanisms.

The impact of the *GABRB3* mutation associated with LGS has also been investigated in mutant, knock-in *Gabrb3^+/D120N^* mice [20]. Like seizures observed in patients carrying the mutation, multiple types of spontaneous seizures including absence, myoclonic, tonic, and generalized tonic-clonic seizures were observed in mice of approximately four months of age. More specifically, the mutant mice showed a seizure frequency of 445 absence, 99 myoclonic, and 4 tonic seizures [20]. As there is a neuropsychiatric comorbidity in LGS, the disease-relevant behavioral abnormalities commonly observed in LGS such as ID and ASD have been evaluated with a battery of behavioral tests conducted in *Gabrb3^+/D120N^* mice. The *Gabrb3^+/D120N^* mice displayed impaired sociability and cognition evaluated with the three-chamber socialization test. In the Barnes maze test, the *Gabrb3^+/D120N^* mice showed a longer latency to find the target hole and increased number of errors in the learning trials. In the memory trial, there was also a significant increase in the number of errors to find the target hole, which both suggests learning and memory deficits in the mice [20].

## 7. Non-GABR Mutations Associated with DEEs and Therapeutic Interventions

Alongside other GABR mutants that contribute to DEE such as GABRA1 [24,51], many other non-GABR ion channel or non-ion channel genes also contribute to DEEs. These genes are involved in kinase activity (cyclin-dependent kinase-like 5) [52], ion channel function (sodium voltage-gated channel alpha subunits 1, 2, and 8; potassium voltage-gated channel subfamily Q member 2) [53,54,55,56], cell adhesion (protocadherin 19) [57], and general cell homeostasis (tuberous sclerosis complex) [58]. There have also been mutations of the GABA transporter (SLC6A1) and previously mentioned NMDA receptors, which attribute to this phenotype [7,9,59,60]. Many genes have been identified as causation for the DEE phenotype, illustrating a need for genetic therapy that can be constructed precisely for each patient (Figure 5) [61,62]. The DEEs are caused by a defective nonfunctional protein due to a genetic mutation; a logical treatment would be therapeutic options through genetic approaches. Antisense oligonucleotide (ASO) therapy is promising, as it provides different methods for treating neurological disorders. For example, the STK-001 ASO encourages exon skipping to increase the amount of productive protein transcript and is currently in phase 1 clinical trial (NCT04442295) for treatment of Dravet syndrome. ASO technology can be employed to treat exonic frameshift or nonsense variants, toxic gain-of-function mutations, and loss-of-function mutations. Although the ethical guidelines for research, development, and use need to be rationed out, this proves to be a promising treatment paradigm moving further into the 21st century [63,64,65].

## 8. Conclusions

In summary, we have reported a case of DEE associated with a deletion in *GABRG2*, which has been previously associated with febrile seizures, CAE, GEFS+, and Dravet syndrome. This is a report of complex movement abnormalities associated with *GABRG2* variants. The seizure phenotype among *GABR* mutations shows overlapping and differential clinical features and, with the genetic mutation being the source of pathophysiology, are defined as developmental and epileptic encephalopathies.

Previous studies indicate mutations in *GABRB3* and *GABRG2* can both give rise to CAE, GEFS+, and DS. This report of DEE associated with *GABRG2* indicates both *GABRB3* and *GABRG2* can give rise to the DEE phenotype. This is consistent with the fact that β_3_ and γ_2_ subunits are partnered in the pentameric receptors at synapses. This is also consistent with the dominant-negative effect displayed by separate mutations of *GABRG2*, which suppress partnering subunits such as *GABRB3*. The deficiency of γ_2_ subunits may compromise the function of partnering subunits such as β_3_ subunits and thus impair the pentameric receptor function. This expands the clinical phenotype spectrum of *GABRG2* mutations in epilepsy and further strengthens the notion of overlapping clinical phenotypes between *GABRG2*, *GABRB3*, and possibly other *GABR* mutations. Alongside other more studied DEE mutations, there is promise in treatment development for DEEs, especially via genetic interventions such as ASOs or gene therapy.

## Figures and Tables

**Figure 1 ijms-23-09683-f001:**
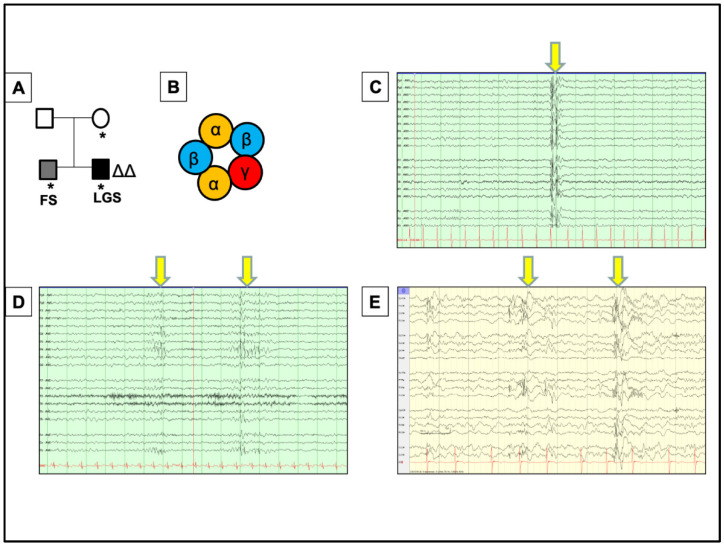
A novel case of Developmental Epileptic Encephalopathy (DEE) associated with *GABRG2* mutation and the patient EEGs. (**A**) A pedigree of seizure disorder identified to be heterozygous deletion of *GABRG2*. * symbolizes *GABRG2* deletion. ΔΔ symbolizes proband, the patient of Developmental Epileptic Encephalopathy (DEE). FS stands for febrile seizures. (**B**) Schematic representation of the major isoform of pentameric GABA_A_ receptor composed of two α, two β, and one γ2 subunit encoded by *GABRG2*. (**C**–**E**) The proband carrying *GABRG2* mutation showed generalized-appearing spike and polyspike wave discharges (**C**); multifocal epileptiform discharges, most prominent in the right mesial parietal (**D**); and left posterior regions (**E**), both during wakefulness and sleep, all obtained around 9 years of age (Appendix A). The image of EEG graph (**C**) is enlarged (Appendix A). The image of EEG graph (**D**) is enlarged (Appendix A). The image of EEG graph (**E**) is enlarged.

**Figure 2 ijms-23-09683-f002:**
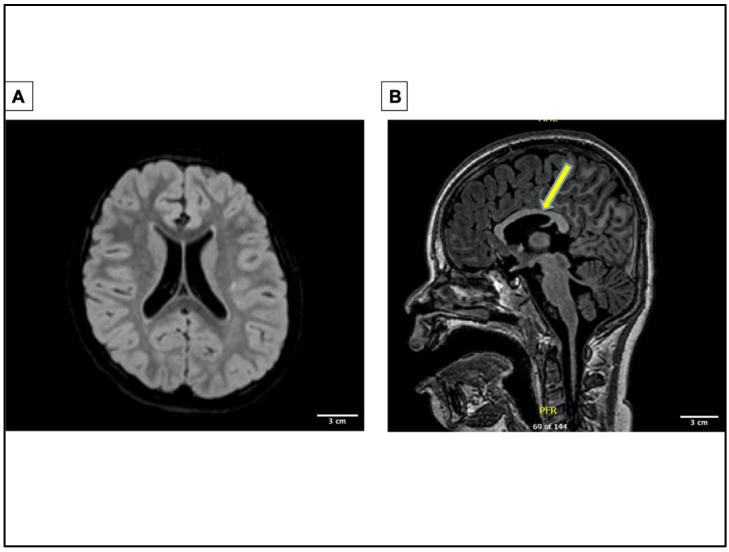
Brain magnetic resonance image of patient showing multifocal abnormality. Multifocal abnormality with abnormal configuration of lateral and third ventricle, suggestion of minimal infra and supratentorial volume loss, microcephaly, abnormal foliar architecture within the cerebellum, and abnormal appearance of the globes. (**A**) MRI image of transverse plane showing abnormal lateral ventricles. (**B**) MRI image of sagittal plane showing abnormal third ventricle indicated by the yellow arrow.

**Figure 3 ijms-23-09683-f003:**
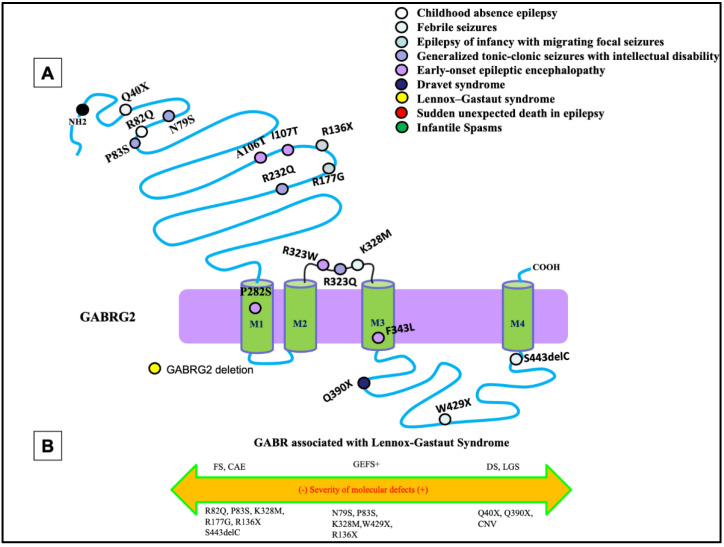
Epilepsy phenotypes associated with *GABRG2* mutations. (**A**) Two-dimensional protein topology of the γ_2_ subunit showing locations of mutations in human *GABRG2* associated with various epilepsy syndromes and neurodevelopmental disorders. Each mutation is color coded to indicate associated diagnosis, and each dot represents the relative location on the γ_2_ subunit protein peptide. (**B**) Diagram dictating the severity of each mutation on a continuum of epilepsy phenotypes. The left side of the diagram infers a less severe epileptic phenotype, the middle dictates a moderate, and the right side of the diagram infers a more severe epileptic phenotype.

**Figure 4 ijms-23-09683-f004:**
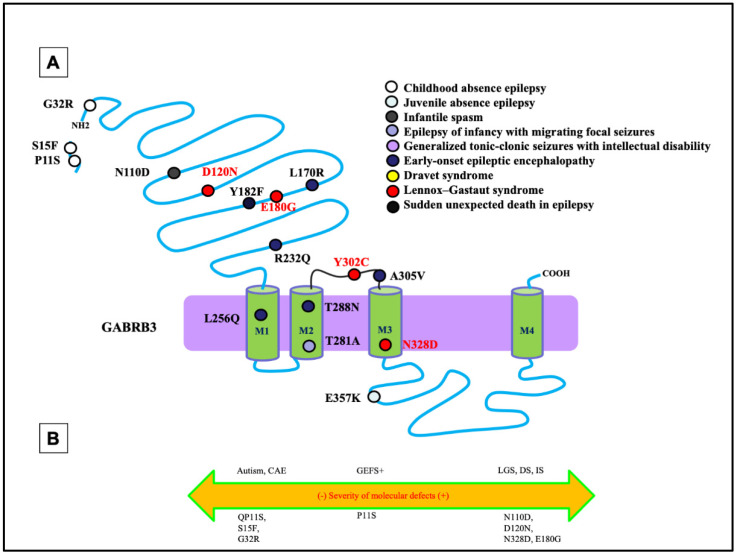
Epilepsy phenotype associated *GABRB3* mutations. (**A**) Two-dimensional protein topology of the β_3_ subunit showing relative locations of mutations in human *GABRB3* associated with various epilepsy syndromes and neurodevelopmental disorders. Each mutation is color coded to indicate associated diagnosis, and each dot represents the β_3_ subunit protein peptide. (**B**) Diagram dictating the severity of each mutation on a continuum of epilepsy phenotypes. The left side of the diagram infers a less severe epileptic phenotype, the middle dictates a moderate, and the right side of the diagram infers a more severe epileptic phenotype.

**Figure 5 ijms-23-09683-f005:**
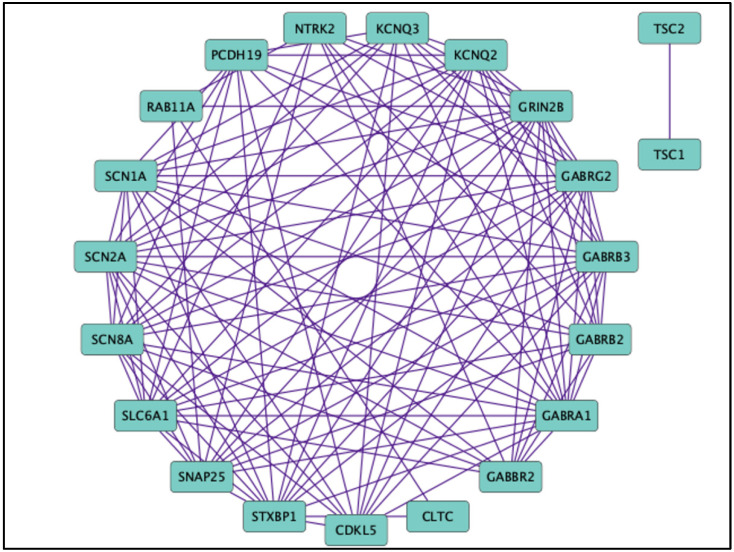
Other genes in addition to GABA_A_ receptor subunit genes associated with developmental epileptic encephalopathies. Protein interaction network displaying genes of varied function discovered to contribute to a diagnosis of DEE. In addition to GABA_A_ receptor subunit genes (GABR), there are other genes unrelated to the GABAergic signaling network, implicating the wide range of pathways contributing to an DEE phenotype. All nodes are grouped by attribute circle layout based on name. Edges are based on publications depicting protein–network interactions. Analysis was generated through String version 11.9 and visualized with Cytoscape version 3.19.

## Data Availability

Not applicable.

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
