# Peer review of "Variable Expression of GABAA Receptor Subunit Gamma 2 Mutation in a Nuclear Family Displaying Developmental and Encephalopathic Phenotype"

_ijms, 2022, doi:10.3390/ijms23179683_

Round 1

Reviewer 1 Report

Mutations in GABAA receptor subunit genes (GABRs) are a major etiology for developmental and epileptic encephalopathies (DEEs). This article reports a nuclear family, consisting of a mother and two twin-brothers who carry a common allelic deletion of the entire coding sequence for the GABRG2 (hemizygous GABRG2), have different phenotypic expressions for DEEs. Authors then compiled an update of molecular mechanisms of GABR mutations, especially the mutations in GABRB3 and GABRG2 attributed to DEEs, with detailed discussion on DEE treatment targets, new treatment paradigms, as well as management for long-term disabilities and improve quality of life for DEE patients. Authors also highlights some potential biomolecular targets for future therapeutic interventions, and genetic approaches a new venue for treatment of DEEs through employing antisense oligonucleotide techniques.

Comments:

1). This is a very interesting case report, with some degree of in-depth current progress review in the field. For the focus and main content of this manuscript, I would suggest author to consider to change the title to “Phenotypic spectrum of Developmental Epileptic Encephalopathies (DEEs) caused by an allelic loss of GABA receptor subunit gamma2 (GABRG2) gene in a nuclear family”.

2). Amazing thing for this family is that all of three sharing the same heterozygous deletion of GABA receptor subunit gamma2 (GABRG2) gene, but with different phenotypes!  As the heterozygous GABRG2 deletion carrier, mother is phenotypically normal, author should discuss, in some detail, whether gender also play a major role for the onset of DEEs.

3). Likewise, the twin-brother of the proband, carrying the same heterozygous GABRG2 deletion, was diagnosed with febrile seizures (FS), which is within the expected range of previous reported phenotypes caused by GABRG2 mutation/deletion, actually, FS is one of the most common phenotypes associated with GABRG2 mutations. Therefore, author need to explore the underlying reason why the proband is differentiated from his twin-brother, presenting with more severe form of DEEs. My suggestion is for a quick genotyping within GABRG2 gene, to evaluate whether the twin-brother inherited two different paternal GABRG2 alleles, which could at least give some indirect explanations for the phenotypic variability for the twin-brother, these genotypic data might also provide supportive foundation for the argument authors made that “Many mutations in non-GABR genes or genes involved in general cell homeostasis have been identified as causation for the DEE phenotype.”

4). In figure 3, authors listed four nonsense mutations (Q40X, R136X, Q390X, and W429X), however, their underlying genetic mechanism are quite different. For example, the phenotypes of first two nonsense mutation are due to genetic term as “hyploinsuffuciency” caused by NMD (nonsense-mediated decay), and the last two nonsense mutation are due to genetic term as “dominant-negative” since both are located in the exon 9 of GABRG2 gene. I would suggest that author should integrate this information into Figure 3.

Therefore, the sentence “The mutant GABRG2 (Q390X) has slow degradation and can impose a dominant-negative effect on the wildtype GABRG2 allele and the partnering subunits (line 199-200).” should be corrected, since there is NO slow degradation for GABRG2 (Q390X) since there is NO NMD for GABRG2 (Q390X) mutant allele. Likewise, the sentence “that generate a premature translation-termination codon and consequently produce a truncated protein [21][22].(line 64-65)” should be corrected as well, since the vast majority of GABRG2 (Q40X) will be degraded due to NMD. Please read the cited references carefully, Reference [22] was trying to rescue the phenotype by suppressing NMD, and promoting reading-through the PTC (premature termination codon) for GABRG2 (Q40X) mutant allele.

Finally, there is no such genetic term as “stop-codon-generating mutations”, I would suggest to change it to “missense, and nonsense (including PTC) mutations”.  Therefore, I strongly suggest that author should consult a human geneticist or scientist with molecular genetics background to review it before send in their revision.

Author Response

Mutations in GABAA receptor subunit genes (GABRs) are a major etiology for developmental and epileptic encephalopathies (DEEs). This article reports a nuclear family, consisting of a mother and two twin-brothers who carry a common allelic deletion of the entire coding sequence for the GABRG2 (hemizygous GABRG2), have different phenotypic expressions for DEEs. Authors then compiled an update of molecular mechanisms of GABR mutations, especially the mutations in GABRB3 and GABRG2 attributed to DEEs, with detailed discussion on DEE treatment targets, new treatment paradigms, as well as management for long-term disabilities and improve quality of life for DEE patients. Authors also highlights some potential biomolecular targets for future therapeutic interventions, and genetic approaches a new venue for treatment of DEEs through employing antisense oligonucleotide techniques.

Comments:

1). This is a very interesting case report, with some degree of in-depth current progress review in the field. For the focus and main content of this manuscript, I would suggest author to consider to change the title to “Phenotypic spectrum of Developmental Epileptic Encephalopathies (DEEs) caused by an allelic loss of GABA receptor subunit gamma2 (GABRG2) gene in a nuclear family”.

  • Reviewers’ Response: We are appreciative of the reviewers well received comment. The proposed title does reflect the wide spectrum of phenotypic expression possible from this mutation. This is a major highlight of the paper and the updated title does reflect the varied expression of GABRG2.

2). Amazing thing for this family is that all of three sharing the same heterozygous deletion of GABA receptor subunit gamma2 (GABRG2) gene, but with different phenotypes!  As the heterozygous GABRG2 deletion carrier, mother is phenotypically normal, author should discuss, in some detail, whether gender also play a major role for the onset of DEEs.

3). Likewise, the twin-brother of the proband, carrying the same heterozygous GABRG2 deletion, was diagnosed with febrile seizures (FS), which is within the expected range of previous reported phenotypes caused by GABRG2 mutation/deletion, actually, FS is one of the most common phenotypes associated with GABRG2 mutations. Therefore, author need to explore the underlying reason why the proband is differentiated from his twin-brother, presenting with more severe form of DEEs. My suggestion is for a quick genotyping within GABRG2 gene, to evaluate whether the twin-brother inherited two different paternal GABRG2 alleles, which could at least give some indirect explanations for the phenotypic variability for the twin-brother, these genotypic data might also provide supportive foundation for the argument authors made that “Many mutations in non-GABR genes or genes involved in general cell homeostasis have been identified as causation for the DEE phenotype.”

  • Reviewers’ Response: We are appreciative of the reviewer’s well received comment. It is important for this paper to explore the phenotypic variability attributed to mutations of GABRG2. Therefore, a compilation of other studied GABRG2 mutations has been included. This alludes to the fact that there is a wide spectrum of clinical diagnosis attributed to mutations of this gene. Your suggestion of genotyping GABRG2 is appreciated as it offers a new avenue for investigation. We have addressed this in the manuscript alluding to the idea that a proper genotype can elucidate the difference in seizure severity. It is very much possible that the mutations are expressed in independent locations on the peptide with variable expression. 

4). In figure 3, authors listed four nonsense mutations (Q40X, R136X, Q390X, and W429X), however, their underlying genetic mechanism are quite different. For example, the phenotypes of first two nonsense mutation are due to genetic term as “hyploinsuffuciency” caused by NMD (nonsense-mediated decay), and the last two nonsense mutation are due to genetic term as “dominant-negative” since both are located in the exon 9 of GABRG2 gene. I would suggest that author should integrate this information into Figure 3.

Therefore, the sentence “The mutant GABRG2 (Q390X) has slow degradation and can impose a dominant-negative effect on the wildtype GABRG2 allele and the partnering subunits (line 199-200).” should be corrected, since there is NO slow degradation for GABRG2 (Q390X) since there is NO NMD for GABRG2 (Q390X) mutant allele. Likewise, the sentence “that generate a premature translation-termination codon and consequently produce a truncated protein [21][22].(line 64-65)” should be corrected as well, since the vast majority of GABRG2 (Q40X) will be degraded due to NMD. Please read the cited references carefully, Reference [22] was trying to rescue the phenotype by suppressing NMD, and promoting reading-through the PTC (premature termination codon) for GABRG2 (Q40X) mutant allele. 

Finally, there is no such genetic term as “stop-codon-generating mutations”, I would suggest to change it to “missense, and nonsense (including PTC) mutations”.  Therefore, I strongly suggest that author should consult a human geneticist or scientist with molecular genetics background to review it before send in their revision.

Reviewer 2 Report

The manuscript entitled “GABAA Receptor Subunit Mutations in Developmental Epileptic Encephalopathies”, by Gerald Nwosu, Shilpa Reddy, Heather Riordan, Jing-Qiong Kang to the journal IJMS as a Review is, in fact, a mixed kind of work, including the description of a clinical case that shows an atypical mutation and an extensive review of the literature without a specific criteria. The authors claim that this is the first case of this kind of mutation (GABRG2) observed in a patient with a Lennox-Gastaut Syndrome (LGS), because the usual mutation is GABRG3. However, there are several and relevant flaws that the authors must address.

 1.- The clinical case is very difficult to follow. The chronology is confusing, describing symptoms at five (line 93), ten (line 138), six (line 141) and four years (line 142). A very relevant information as the treatment of a hepatoblastoma (what age?) is offered in lines 105-107. Therefore, it is needed a complete re-elaboration of the clinical case following the classical principles of Medicine: familial and personal antecedents, symptomatology and exploration, accessory studies (analytical, EEG, imaging), evolution and treatment. Please, clarify what means that the proband is working with a communication device (line 144). It is extremely significant to describe in detail the next questions:

1.1.- The seizures semiology, because the LGS (see below) is highly associated with some kinds of seizures. The authors only state that seizures show multiple semiology (line 94) but they need describe in detail this semiology.

1.2.- How do the authors were sure that myoclonus was non-epileptic (line 101, 139)?. They must show a simultaneous EEG + EMG recording during myoclonus to show that there is no irritative activity associated. This affirmation is difficult to combine with “The background activity was consistent with (…) myoclonic seizures” (lines 94-96)?. In this sense, a background consistent with does not prove neither excludes these types of seizures.

1.3.- The authors should show ictal EEG recordings, or better, video-electroencephalography (vEEG). This is a very important point because irritative activity showed by the authors is highly unspecific and compatible with some other DEEs. In fact, there are two types of recordings that the authors must show to convince readers that this is a LGS: slow spike-wave (around 2.5 Hz) and burst of fast activity during NREM sleep.

 1.4.- The authors affirm that they have made the “diagnosis of LGS (…) not based upon EEG findings as these can vary between patients and over time within the same patient” (lines 147-149). However, this sentence is highly dubious. Next I enlist three definitions of LGS from authorities (ILAE) or experts in epilepsy. The EEG criteria for definition are underlined:

a)     This syndrome is characterized by the presence multiple types of intractable seizures (in particular tonic seizures in sleep, but atonic and atypical absence seizures also occur), cognitive and behavioral impairments and diffuse slow spike-and-wave and paroxysms of fast activity on EEG. International League Against Epilepsy (https://www.epilepsydiagnosis.org/syndrome/lgs-overview.html, visited August, 2nd of 2022)

b)    “Tonic seizures are the characteristic and defining seizure type in the Lennox–Gastaut syndrome, and this is their usual clinical setting” (p. 9); [atonic seizures] “are common in severe symptomatic epilepsies (especially in the Lennox–Gastaut syndrome)” (p. 10) “Characteristic EEG pattern—slow spike-wave (≤ 2.5 Hz), abnormal background, bursts of fast (≥ 10 Hz) activity in non-REM sleep” (p. 23), Handbook of epilepsy treatment. Simon D Shorvon, 2nd Ed. Blackwell, 2005.

c)     “LGS is defined by several criteria: 1. polymorphous epileptic seizures, with mainly atypical absences, axial tonic and atonic seizures (tonic seizures, atypical absences and drop attack); 2. EEG patterns consisting of diffuse slow SWs and bursts of fast rhythms at 10–12 Hz during sleep; and 3. Permanent psychological disturbances with psychomotor delay or personality disorders or both.” Textbook of Epilepsy Surgery, Ed. H.O Lüders, Informa Health, 2008; p. 384

d)    “The outstanding feature is the slow spike-wave complex ranging from 1 to 2.5 Hz. It is more often an interictal rather than an ictal discharge, is most often seen in a generalized synchronous pattern, although lateralization is also fairly common. Focal slow spike-wave activity is quite rare (…) another important pattern is runs of rapid spikes, which are seen in non-REM sleep only. This pattern is more common in older children, adolescents, or adults” Neidermeyer’s Electroencephalography. 6th Ed. D.L. Schomer, F. H. Lopes Da Silva. Wolters Kluwer/Lippincot Williams & Wilkins, 2011; pp: 519-520.

By resume, the main finding of the case, i.e a new mutation in LGS other than GABRG3 fails if the authors do not prove that this patient, in fact suffers a LGS. The case do not fit well into the description of LGS, except for the presence of cognitive and behavioural impairment. Therefore, if they do not prove this diagnosis, the manuscript loss its meaning.

 2.- Regarding the hepatoblastoma, please explain how the chemotherapy can change the gene expression (epigenetics?) and offer bibliography. This chemotherapy, was continued or finished at any time?. If finished, can we expect that epileptogenic genes would express again changing the phenotype?.

 3.- Figure 2A and 2B are poorly informative. Can be removed and use the space to show better the EEG recordings.

 4.- Figure 5. Please, clarify what nodes you are referring and what is the meaning of this figure.

 5.- The second paragraph of Conclusion seems more a Discussion.

Other minor comments

1.- I’m not sure that the topographic descriptor “mesial” (line 117) would be appropriate, because this word usually refers to structures near the midline.

 2.- Use the singular to refer to the third ventricle (line 137)

 3.- Define ASD the first time it appears (line 141)

 4.- Please, clarify the difference between GABA potency and GABA efficacy (lines269-270)

 5.- Please, check lines 301-302 regarding the expression 445 absence, ninety-nine myoclonic and four tonic. Are you referring to seizure frequency?

Author Response

The manuscript entitled “GABAA Receptor Subunit Mutations in Developmental Epileptic Encephalopathies”, by Gerald Nwosu, Shilpa Reddy, Heather Riordan, Jing-Qiong Kang to the journal IJMS as a Review is, in fact, a mixed kind of work, including the description of a clinical case that shows an atypical mutation and an extensive review of the literature without a specific criterion. The authors claim that this is the first case of this kind of mutation (GABRG2) observed in a patient with a Lennox-Gastaut Syndrome (LGS), because the usual mutation is GABRG3. However, there are several and relevant flaws that the authors must address.

 1.- The clinical case is very difficult to follow. The chronology is confusing, describing symptoms at five (line 93), ten (line 138), six (line 141) and four years (line 142). A very relevant information as the treatment of a hepatoblastoma (what age?) is offered in lines 105-107. Therefore, it is needed a complete re-elaboration of the clinical case following the classical principles of Medicine: familial and personal antecedents, symptomatology and exploration, accessory studies (analytical, EEG, imaging), evolution and treatment. Please, clarify what means that the proband is working with a communication device (line 144). It is extremely significant to describe in detail the next questions:

1.1.- The seizures semiology, because the LGS (see below) is highly associated with some kinds of seizures. The authors only state that seizures show multiple semiology (line 94) but they need describe in detail this semiology.

1.2.- How do the authors were sure that myoclonus was non-epileptic (line 101, 139)?. They must show a simultaneous EEG + EMG recording during myoclonus to show that there is no irritative activity associated. This affirmation is difficult to combine with “The background activity was consistent with (…) myoclonic seizures” (lines 94-96)?. In this sense, a background consistent with does not prove neither excludes these types of seizures.

1.3.- The authors should show ictal EEG recordings, or better, video-electroencephalography (vEEG). This is a very important point because irritative activity showed by the authors is highly unspecific and compatible with some other DEEs. In fact, there are two types of recordings that the authors must show to convince readers that this is a LGS: slow spike-wave (around 2.5 Hz) and burst of fast activity during NREM sleep.

 1.4.- The authors affirm that they have made the “diagnosis of LGS (…) not based upon EEG findings as these can vary between patients and over time within the same patient” (lines 147-149). However, this sentence is highly dubious. Next I enlist three definitions of LGS from authorities (ILAE) or experts in epilepsy. The EEG criteria for definition are underlined:

  1. a) This syndrome is characterized by the presence multiple types of intractable seizures (in particular tonic seizures in sleep, but atonic and atypical absence seizures also occur), cognitive and behavioral impairments and diffuse slow spike-and-wave and paroxysms of fast activity on EEG. International League Against Epilepsy (https://www.epilepsydiagnosis.org/syndrome/lgs-overview.html, visited August, 2nd of 2022)
  2. b)“Tonic seizures are the characteristic and defining seizure type in the Lennox–Gastaut syndrome, and this is their usual clinical setting” (p. 9); [atonic seizures] “are common in severe symptomatic epilepsies (especially in the Lennox–Gastaut syndrome)” (p. 10) “Characteristic EEG pattern—slow spike-wave (≤ 2.5 Hz), abnormal background, bursts of fast (≥ 10 Hz) activity in non-REM sleep” (p. 23), Handbook of epilepsy treatment. Simon D Shorvon, 2nd Ed. Blackwell, 2005. 
  3. c)“LGS is defined by several criteria: 1. polymorphous epileptic seizures, with mainly atypical absences, axial tonic and atonic seizures (tonic seizures, atypical absences and drop attack); 2. EEG patterns consisting of diffuse slow SWs and bursts of fast rhythms at 10–12 Hz during sleep; and 3. Permanent psychological disturbances with psychomotor delay or personality disorders or both.” Textbook of Epilepsy Surgery, Ed. H.O Lüders, Informa Health, 2008; p. 384
  4. d)“The outstanding feature is the slow spike-wave complex ranging from 1 to 2.5 Hz. It is more often an interictal rather than an ictal discharge, is most often seen in a generalized synchronous pattern, although lateralization is also fairly common. Focal slow spike-wave activity is quite rare (…) another important pattern is runs of rapid spikes, which are seen in non-REM sleep only. This pattern is more common in older children, adolescents, or adults” Neidermeyer’s Electroencephalography. 6th D.L. Schomer, F. H. Lopes Da Silva. Wolters Kluwer/Lippincot Williams & Wilkins, 2011; pp: 519-520.

By resume, the main finding of the case, i.e a new mutation in LGS other than GABRG3 fails if the authors do not prove that this patient, in fact suffers a LGS. The case do not fit well into the description of LGS, except for the presence of cognitive and behavioural impairment. Therefore, if they do not prove this diagnosis, the manuscript loss its meaning.

  • Reviewers’ Response: The reviewer’s comments were well received and appreciated. The manuscript has been updated to follow the classical principles of medicine as suggested. This is most appreciated as it allows clinicians and researchers alike to clearly interpret the case report. Clinical information is limited on the patient so the line describing the patients communication device has been removed. As well we can not provide simultaneous EEG and EMG recordings to display whether myoclonus is non-epileptic. The reported observations is based upon familial observations noted in the patient report. Last, the diagnosis of LGS has been removed from this paper. The patient report received from clinicians does not provide enough information for a proper diagnosis of LGS. However, the proband does still display developmental and epileptic phenotypes indicative of DEEs

 2.- Regarding the hepatoblastoma, please explain how the chemotherapy can change the gene expression (epigenetics?) and offer bibliography. This chemotherapy, was continued or finished at any time?. If finished, can we expect that epileptogenic genes would express again changing the phenotype?.

 3.- Figure 2A and 2B are poorly informative. Can be removed and use the space to show better the EEG recordings.

 4.- Figure 5. Please, clarify what nodes you are referring and what is the meaning of this figure.

 5.- The second paragraph of Conclusion seems more a Discussion.

Other minor comments

1.- I’m not sure that the topographic descriptor “mesial” (line 117) would be appropriate, because this word usually refers to structures near the midline. 

 2.- Use the singular to refer to the third ventricle (line 137)

 3.- Define ASD the first time it appears (line 141)

 4.- Please, clarify the difference between GABA potency and GABA efficacy (lines269-270)

 5.- Please, check lines 301-302 regarding the expression 445 absence, ninety-nine myoclonic and four tonic. Are you referring to seizure frequency?

  • Reviewers’ response: All minor comments have been addressed in the text.